# Performance Evaluation of a Voice-Based Depression Assessment System Considering the Number and Type of Input Utterances

**DOI:** 10.3390/s22010067

**Published:** 2021-12-23

**Authors:** Masakazu Higuchi, Noriaki Sonota, Mitsuteru Nakamura, Kenji Miyazaki, Shuji Shinohara, Yasuhiro Omiya, Takeshi Takano, Shunji Mitsuyoshi, Shinichi Tokuno

**Affiliations:** 1Department of Bioengineering, Graduate School of Engineering, The University of Tokyo, Tokyo 113-8656, Japan; noriaki.sonota@gmail.com (N.S.); nakamura@bioeng.t.u-tokyo.ac.jp (M.N.); shinohara@bioeng.t.u-tokyo.ac.jp (S.S.); mitsuyoshi@bioeng.t.u-tokyo.ac.jp (S.M.); or shi.tokuno.shinichi@gmail.com (S.T.); 2Mitsui Knowledge Industry Co., Ltd., Tokyo 105-6215, Japan; miyazaki-kenji@mki.co.jp; 3PST Inc., Kanagawa 231-0023, Japan; omiya@medical-pst.com (Y.O.); takano@medical-pst.com (T.T.); 4Graduate School of Health Innovation, Kanagawa University of Human Services, Kanagawa 210-0821, Japan

**Keywords:** mental health, voice biomarker, mood change by positive utterances

## Abstract

It is empirically known that mood changes affect facial expressions and voices. In this study, the authors have focused on the voice to develop a method for estimating depression in individuals from their voices. A short input voice is ideal for applying the proposed method to a wide range of applications. Therefore, we evaluated this method using multiple input utterances while assuming a unit utterance input. The experimental results revealed that depressive states could be estimated with sufficient accuracy using the smallest number of utterances when positive utterances were included in three to four input utterances.

## 1. Introduction

Mental health care has become increasingly important worldwide due to stressful conditions in current society. Stress has a negative effect on health and mood in daily life, and accumulating stress induces mental and behavioral disorders [1]. These disorders cause lifetime income loss and decreased labor productivity, thereby resulting in a substantial economic loss to society [2,3]. Since early diagnosis and interventions for depression can result in a high remission rate [4], technologies that can easily detect early-stage depressive symptoms, stress states, and mental disorders are essential.

Although screening methods are available for diagnosing patients with mental illness using biomarkers such as saliva and blood [5,6], they are invasive and require special measuring instruments and devices, thereby increasing cost. Commonly used non-invasive methods include self-administered psychological tests, such as the Patient Health Questionnaire 9 (PHQ9), General Health Questionnaire (GHQ), and Beck Depression Inventory (BDI) [7,8,9]. Although these self-administered psychological tests are relatively simple to implement, they cannot rule out the drawback of reporting biases [10] wherein conscious and subconscious decisions of the respondent result in specific information being selectively underestimated or overestimated. Physician-based examinations such as the Hamilton Depression Rating Scale (HDRS/HAM-D) [11] are available as well; however, they are time consuming and limited by the number of examinations that the physicians can implement.

It is empirically and academically known that changes in mood appear in facial expressions and voices [12,13], and research has been conducted to employ these aspects for estimating depressive or stressed states [14,15,16,17]. Among these, voice-based analyses are considered advantageous as they are non-invasive and can be conducted remotely without any specialized equipment. In addition, they can potentially resolve reporting biases during self-administered psychological tests and detect various psychiatric disorders. As a result, it has been attracting attention in recent years.

The authors previously developed an index of “vitality” that expresses the degree of mental health using voices [18] and distributed it using software development kits (SDKs) and smartphone/online applications (Mind Monitoring System, MIMOSYS) [19], aiming wide availability to multiple companies and general users. This method could monitor the degree of mental health on a daily basis and prevent mental disorders such as depression. It employed voices on daily telephone calls since the natural utterance in a telephonic call voice includes multiple types of utterances. Herein, the vitality for each utterance was calculated and averaged; thereafter, it was applied as the final vitality of the input voice. If users experienced difficulty in using the telephone, they could still use the method by inputting various routine phrases in a pseudo-call state. To obtain stable results, a certain number of utterances were required; empirical results suggested that at least six utterances in the voice input were required to achieve sufficient accuracy. However, in daily measurements, reducing the number of input utterances to its lowest value would be ideal for applications such as continuous monitoring of stress in companies.

Mood-based research often involves artificially inducing a specific mood in participants. One such inducing method is verbal instruction wherein the participants are asked to read a positive sentence to induce a pleasant mood and a negative sentence to induce an unpleasant mood [20], thereby suggesting that the mood of the reader changes depending on the meaning of the sentence being read. However, it is thought that individuals suffering from depression have poor emotional expression and minimal mood swings; therefore, applying the abovementioned method to induce pleasant or unpleasant feelings may result in some differences in vitality among depressive groups when compared to healthy groups. Confirming any characteristic tendencies in those differences may lead to a reduction in the necessary number of input utterances.

Therefore, to achieve an accurate estimate of depressive states with a minimal number of input utterances, we conducted an assessment evaluation of the abovementioned method for the number of input utterances among self-reported healthy subjects and physician-diagnosed depressive subjects, while considering semantic differences in their utterances.

## 2. Materials and Methods

### 2.1. Ethical Considerations

This study was conducted with the approval of the Ethical Review Board of the University of Tokyo and National Defense Medical College (No. 11572, No. 2248).

### 2.2. Subjects

The total of 130 subjects were recruited from two hospitals, namely the National Defense Medical College Hospital (H1), and the Tokyo Medical University Hospital (H2).

At H1, patients undergoing treatment for major depressive disorder were informed about our study at their first visit to the hospital. Then, 93 patients who gave informed consent were collected. In addition, 14 self-reported healthy volunteers (colleagues and university staff) were recruited and their informed consent for our study was obtained. At this hospital, physicians performed HDRS on depressed patients to assess their severity. Thereafter, in accordance with the 21-item version of the assessment criteria for HDRS reported by a document [21], subjects with an HDRS score of less than five were treated as healthy subjects. At H2, a total of 23 subjects (colleagues and connections) were recruited, all of them self-reported as healthy and their informed consent for our study was obtained.

Based on the abovementioned assessment criteria, 92 and 38 subjects were assigned to the healthy group and the depressive group, respectively. The detailed information of the subjects is presented in Table 1.

### 2.3. Voice Recordings

Voice recordings were conducted at the examination rooms of the individual hospitals, and subjects were requested to read 17 routine phrases in Japanese. Of the 17 phrases, 11 did not include emotions, four included positive emotions (P1–P4), and two included negative emotions (N1–N2). The phrases were selected in consultation with a psychiatrist. In particular, the positive and negative phrases were chosen from the items that psychiatrists use in their daily interviews with patients. In this study, the number of positive, negative, and other routine phrases were equalized to the maximum extent by selecting four out of the 11 emotionless phrases (E1–E4) as the other phrases, and a total of 10 routine phrases were analyzed. The contents of the routine phrases to be analyzed are presented in Table 2.

### 2.4. Voice Analysis

Voice analysis was conducted based on the sensibility technology (ST) [22]. ST analyzes change patterns of fundamental frequencies from voices and calculates the extent of the emotions included in the voice, namely “calm”, “anger”, “joy”, “sorrow”, and “excitement.” The method evaluated in this study calculates “vitality” that quantifies the extent of the mental health of the user from the emotions analyzed by ST. Vitality can have any value between 0 to 1, and a good mental health value lies within the range of approximately 0.55 ± 0.1 [19,23]. This is because some mental abnormalities have been determined for extremely high or low values. In our previous studies, a correlation between BDI score and vitality [24] has been reported. In addition, a correlation between vitality and the HDRS for individuals with depression [25] has been reported.

“Utterance” is the smallest unit of speech analyzed by ST and is a continuous voice divided by breathing, pauses, etc. In this study, the start of an utterance was detected when the silent state changed to a vocalized state and was continued for a certain period of time. The end of the utterance was detected when the vocalized state changed to the silent state for a certain period of time. A given state was determined as silent or vocalized by setting a threshold amplitude value for the time waveform of the voice. The vitality of each utterance in the voice was calculated and averaged to obtain an output vitality for that voice.

Each routine phrase read by the subject was regarded as a single utterance, and the average of the vitality calculated for each routine phrase was set as the standard vitality for the individual subjects. In order to show that vitality is a valid index to assess depression, the correlation between HDRS score and vitality in depressed patients was analyzed. As voice data recorded in multiple facilities were analyzed, the differences in vitality were measured as per the recording location. The average value for the standard vitality in the healthy group was compared across both facilities. In addition, since the age of the subjects was distributed across a wide range, age-based differences in vitality were analyzed as well. A combination of the 10 obtained utterances were considered for analyzing the number of utterances, and the vitality for each individual utterance was averaged. If the number of utterances was *k*, ${}_{10}C_{k}$ combinations of utterances were obtained. Thereafter, the vitality average for each subject was determined for each combination. A receiver operating characteristic (ROC) curve was considered as the discrimination index between the healthy group and depressive group, and the area under the curve (AUC) was employed as the discrimination accuracy. These calculations resulted in ${}_{10}C_{k}$ AUC values, using which the maximum, minimum, and average values were calculated. In addition, the absolute difference (absolute value difference) between the vitality average and the standard vitality was calculated for each combination, averaged among the subjects, and set as the difference from the standard vitality. A total of ${}_{10}C_{k}$ values were obtained for the differences as well, and their maximum, minimum, and average values were calculated. The details of these calculations are displayed in Figure 1. R version 4.0.2 [26], a free software for statistical analysis, was used for statistical processing.

## 3. Results

### 3.1. Validity of Vitality

Figure 2 displays the scatter plot of HDRS score and standard vitality in depressed patients. A significant weak correlation was obtained (r=−0.21,p=0.048).

### 3.2. Differences Due to Recording Location

Since vitality follows a normal distribution [23], we applied parametric statistical methods. First, the homoscedasticity of the standard vitality was examined between the facilities using the F-test. The results revealed that for the healthy group, the standard variance was not significantly different across recording locations (F(22,68)=1.22,p=0.52). Next, the results of comparing the average value of standard vitality between facilities using the *t*-test revealed no significant differences (t(90)=0.96,p=0.34).

### 3.3. Differences Due Age

Figure 3 displays the scatter plots of the age and standard vitality of the subjects in the healthy and depressive groups. As per the figure, no significant correlations were obtained in the healthy group (r=−0.083,p=0.45) and the depressive group (r=−0.0067,p=0.97). Therefore, there was no correlation between age and standard vitality.

On the other hand, a significant difference was found when comparing the average value of standard vitality between the healthy group and the depressive group using the *t*-test (t(122)=3.09,p=0.0025). Then, the standard vitality was adjusted for age using analysis of covariance, and the average value of adjusted standard vitality was compared between the healthy group and the depressive group using *t*-test, and a significant difference was found (t(122)=2.97,p=0.0036). Figure 4 displays the box plots of the standard and adjusted standard vitality distributions for the healthy and depressive groups.

### 3.4. Differences Due to Number of Utterances

The results of healthy group/depressive group discrimination accuracy for the number of utterances k=(1,2,…,10) are displayed in Figure 5. In the figure, the green line represents the AUC for standard vitality, and the red and blue dots respectively represent the maximum and minimum AUC values of the AUCs obtained for the vitality average by combining the unit utterances for each number of utterances. The black dots represent the AUC average, and the upper and lower bars represent the standard error. Table 3 presents the combinations of utterances that provided the maximum/minimum AUC for up to six utterances.

The results obtained for analyzing the differences from standard vitality for the number of utterances k=(1,2,…,10) are displayed in Figure 6. The red and blue dots respectively represent the largest and smallest differences in the obtained difference from standard vitality. The green dots represent differences obtained from the combination of utterances that gave the maximum AUC. The black dots represent the average of the difference, and the upper and lower bars represent the standard error. In addition, the combinations of utterances that gave the maximum/minimum differences for up to six utterances are presented in Table 4.

### 3.5. Differences Due to Type of Utterance

A vitality distribution divided between the healthy group and the depressive group for each utterance is illustrated in Figure 7.

Utterances where significant differences were confirmed by the *t*-test in the vitality average between the healthy group and the depressive group were P3 (t(128)=2.10,p=0.038), P4 (t(128)=2.06,p=0.041), E2 (t(127)=2.61,p=0.010), and E3 (t(127)=2.32,p=0.022).

## 4. Discussion

Based on our previous studies and the correlation between HDRS score and standard vitality in depressed patients in this study, vitality was considered to be a valid index to assess depression. Since the number of severely depressed patients in this study was small, it was considered that the correlation between HDRS score and standard vitality in depressed patients was not sufficient.

Since the recordings were conducted at multiple facilities, we evaluated whether any potential difference was caused by the recording location; however, no significant differences were found in the standard vitality of the healthy group. Therefore, the location of the recording was not taken into consideration.

As the ages of the subjects were distributed across a wide range, any potential difference due to age was evaluated as well; however, no significant correlations were found between the standard vitality and age of both the healthy group and the depressive group. In addition, for both the healthy and depressive groups, the standard vitality distributions were almost the same before and after adjustment for age. Therefore, age was not considered as a potential evaluation parameter for the experimental data. We recruited subjects from a wide range of age groups, including confirmation that there was no effect of age on the vitality. In fact, it was confirmed that the analysis result did not change with the standard vitality adjusted for age.

The significant difference in the standard vitality between the healthy group and the depressive group may have potential factors other than the recording environment and age; however, further analysis of the factors is not possible in this study and is a future study.

Based on the transitions of the average AUC values, the discrimination accuracy between the healthy group and the depressive group tended to increase as the number of utterances increased. This was consistent with the previous internal verification results of the authors (unpublished as a paper). The maximum AUC value tended to increase when the number of utterances was low and reached a maximum level when there were four utterances, thereby reaching a value that was higher than the AUC value obtained by standard vitality; the value showed a decreasing tendency with subsequent utterances. Interestingly, there was a tendency for multiple positive utterances to be included in the combinations of utterances that gave a maximum AUC value. Conversely, there was a tendency for multiple negative utterances to be included in combinations of utterances that gave a minimum AUC value. As the number of included negative utterances was lower than the positive utterances among the read utterances, this tendency may seem obvious. However, upon reviewing the utterance content, there was at least one positive utterance in all utterances combinations that gave the maximum AUC value for up to six utterances (except for when the number of utterances was one). In addition, herein, the average positive utterance inclusion rate was approximately 43%, the average negative utterance inclusion rate was 0% (not included), and the average other utterance inclusion rate was approximately 57%. Meanwhile, at least half of the negative utterances were included in all utterance combinations that gave the minimum AUC value for up to six utterances. Herein, the average negative, positive, and other utterance inclusion rate was approximately 57%, 9%, and 34%, respectively. Despite the fact that there were only two negative utterances in the read utterances, the negative utterance inclusion rate tended to exceed that of the positive utterance inclusion rate. This could be caused by the change in the mood of the reader due to the meaning of the read utterance. In other words, since the healthy group exhibited a rich emotional expression, their mood was elevated and they felt energized while reading positive utterances; in contrast, since the depressive group portrayed poor emotional expression, they exhibited no particular mood change even while reading positive utterances. Therefore, reading multiple positive utterances could clarify the difference in vitality between the healthy group and the depressive group. Conversely, the healthy group felt discontent and less energetic upon reading a negative utterance, reaching a similar vitality level as that of the depressive group; in contrast, the depressive group did not exhibit a substantial mood change. Therefore, the differences in vitality between the two groups became more challenging to interpret. To confirm this result, we investigated the vitality distributions for each type of utterance. These distributions are for the vitality of one utterance, so each value is unstable; nevertheless, we observed differences in the distributions. For utterances P3 and P4, the healthy group had a slight tendency to become more energetic, whereas the depressive group had a slight tendency to become less energetic; therefore, the difference in the vitality distribution was significant. For utterances N1 and N2, no significant differences could be confirmed in the vitality distributions of the healthy group and the depressive group; therefore, suggesting that both groups were at the same level. However, for N1, the distribution of the healthy group and the depressive group both tended to be higher than normal; therefore, the groups did not always become less energetic with a negative utterance. For the other utterances E3 and E4, although the vitality tended to be relatively higher, the reason for this result was unclear. For E2 and E3, although significant differences could be confirmed in the vitality distributions of the healthy group and the depressive group, the corresponding reasons were unclear as well. The obtained difference could be due to the fact that E2 and/or E3 were included for each number of utterances among the combinations of utterances that gave the maximum AUC value.

Differences from the standard vitality tended to decrease as the number of utterances increased; this is a natural tendency. However, for utterance combinations that gave the maximum/minimum difference, multiple negative utterances were not always included when the difference was at a maximum, and vice versa. The healthy group/depressive group discrimination accuracy is important for vitality; therefore, we evaluated the differences in the combinations of utterances that gave the maximum AUC value. The obtained results were consistent with the minimum difference values when there were one or two utterances, and close to the average difference for higher numbers of utterances. When there were four utterances, the difference was significant high at approximately 0.07 (t(129)=5.60,p<0.01). However, vitality is inherently an index with a large variation, and standard vitality may not always provide the correct value for a subject; therefore, it may not be appropriate to make certain conclusions based on the differences obtained from this standard value.

From the above, it was concluded that upon considering reductions from the six utterances proposed by the authors, it is desirable to include the positive utterances P3 (“I have an appetite”) and P4 (“I feel calm”) among the four utterances to be included while considering discrimination accuracy between the healthy group and the depressive group. The combination of three utterances that gave a maximum AUC value included P4 as well, and although the discrimination accuracy was somewhat inferior to that of four utterances, it could be ignored for practical applications. Validation with other datasets is needed to ensure the reliability and validity of our results, which is a topic of future study.

## 5. Conclusions

In this study, in order to accurately estimate the depressive state with a small number of input utterances using a voice-based method developed by us, we assessed its performance with respect to the number of utterances on self-reported healthy subjects and physician-diagnosed depressive subjects, while considering semantic differences in utterances. The experimental results suggested that depressive states may be estimated with sufficient accuracy with the lowest number of utterances when utterances with a positive meaning were included in three to four utterances.

Topics of future research include evaluating specific types of positive utterances for effective discrimination since some utterances did not contribute to the discrimination of depressive states despite intending a positive meaning. In addition, the reported method calculates an additional index called “mental activity” along with vitality. This index is obtained by taking a moving average of the vitality obtained using sequential data and has a higher accuracy than vitality. The discrimination accuracy obtained by continuously inputting multiple utterances was assessed in this study. However, when multiple utterances are input over certain time intervals and a similar assessment is conducted, one utterance a day could achieved a similar level of discrimination accuracy as that of the mental activity; thereby increasing the convenience of the reported method. Therefore, assessing the discrimination accuracy over certain time intervals is an important topic of future study as well.

## Figures and Tables

**Figure 1 sensors-22-00067-f001:**
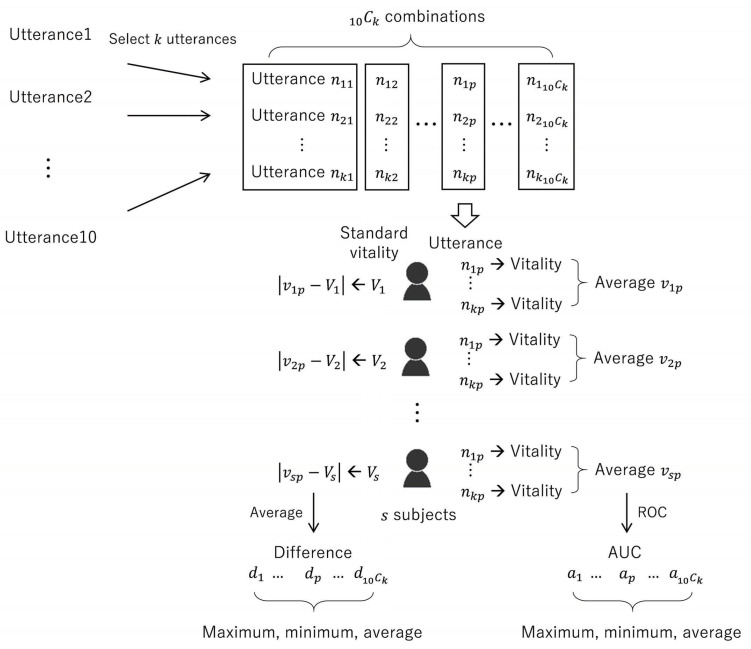
Details of utterance analysis.

**Figure 2 sensors-22-00067-f002:**
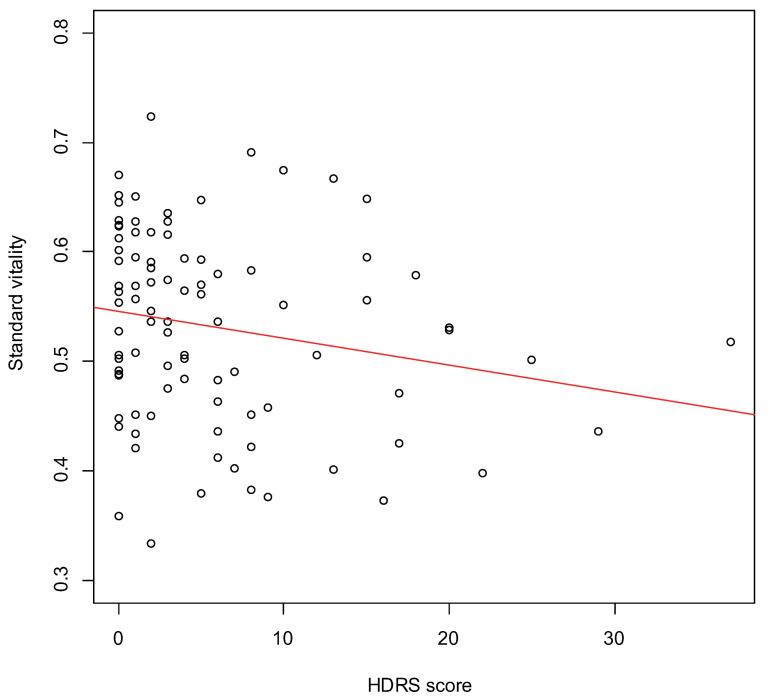
The scatter plot of HDRS score and standard vitality in depressed patients.

**Figure 3 sensors-22-00067-f003:**
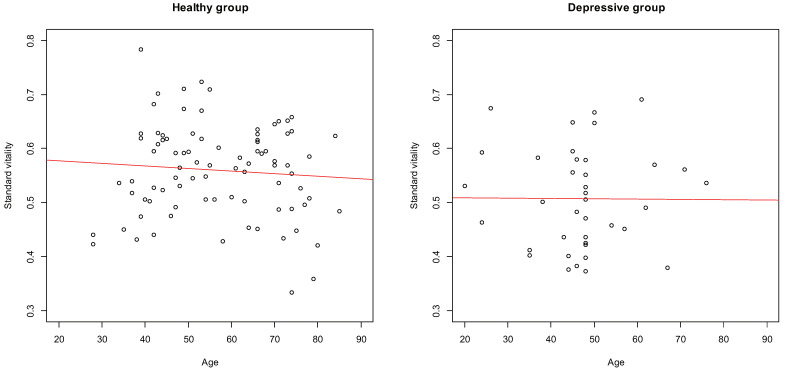
Scatter plots of age and standard vitality.

**Figure 4 sensors-22-00067-f004:**
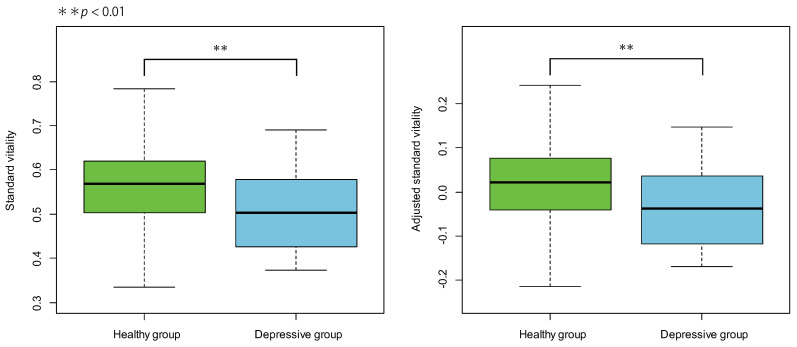
Box plots of the standard and adjusted standard vitality distributions for the healthy and depressive groups.

**Figure 5 sensors-22-00067-f005:**
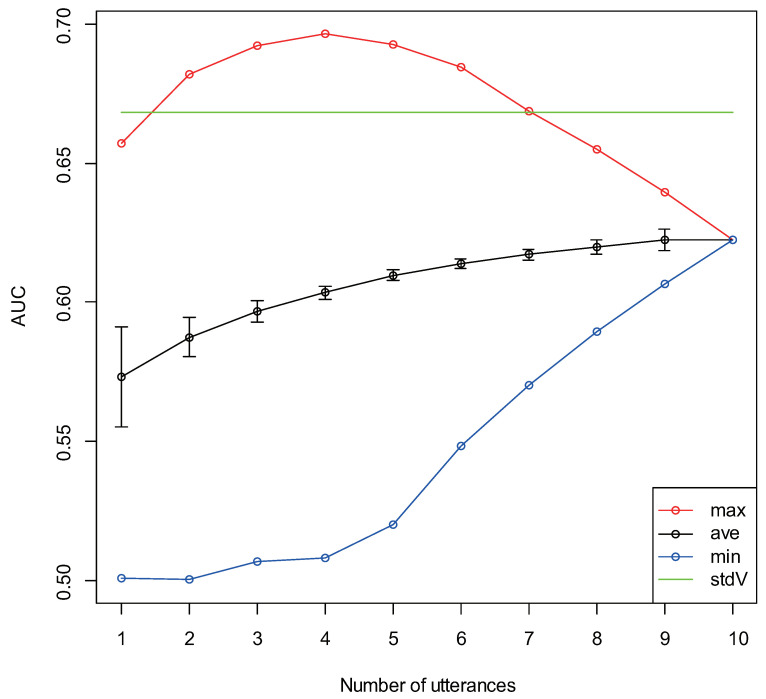
Healthy group/depressive group discrimination accuracy for each number of utterances.

**Figure 6 sensors-22-00067-f006:**
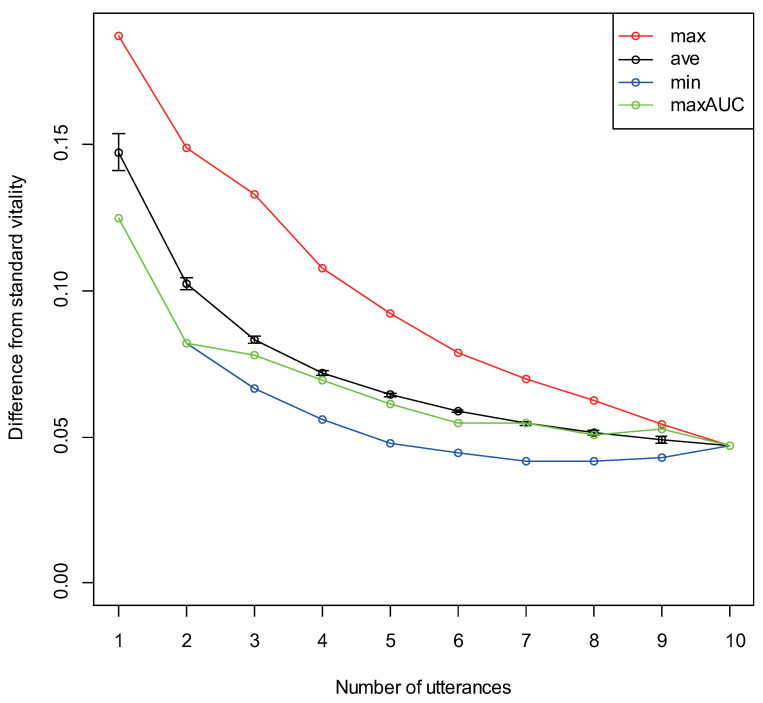
Difference from standard vitality for each number of utterances.

**Figure 7 sensors-22-00067-f007:**
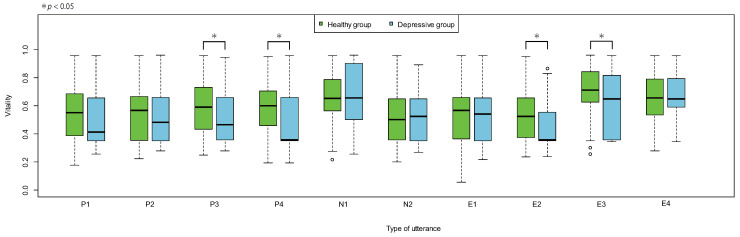
Vitality distribution of healthy group/depressive group by utterance.

**Table 1 sensors-22-00067-t001:** Detailed information of subjects.

Facility	Sex	Number of Healthy Subjects	Number of Depressed Patients		Age (Mean ± SD)
			HDRS < 5	HDRS ≥ 5	
H1	Male	10	21	27	48.3 ± 12.9
	Female	4	34	11	60.7 ± 14.3
	Total	14	55	38	53.9 ± 14.8
H2	Male	12	0	0	47.6 ± 10.6
	Female	11	0	0	59.8 ± 13.3
	Total	23	0	0	53.4 ± 13.2
		Healthy group total 92		Depressive group total 38	

**Table 2 sensors-22-00067-t002:** Content of routine phrases.

Type	Phrase
P1	I’m feeling very well
P2	I slept very well yesterday
P3	I have an appetite
P4	I feel calm
N1	I am tired and drained
N2	I am short-tempered
E1	I-ro-ha-ni-ho-he-to (Similar to “a–b–c”)
E2	It’s fine today (Mic test phrase commonly used in Japan)
E3	Once upon a time
E4	Galapagos Islands

**Table 3 sensors-22-00067-t003:** Combinations of utterances that gave the maximum/minimum AUC for each number of utterances (up to six utterances).

Number of Utterances	Combination of Utterances That Gives Maximum AUC	Combination of Utterances That Gives Minimum AUC
1	E2	N2
2	P4, E2	N1, E4
3	P4, E2, E3	N1, N2, E4
4	P3, P4, E2, E3	N1, N2, E1, E4
5	P1, P3, P4, E2, E3	P2, N1, N2, E1, E4
6	P1, P2, P3, P4, E2, E3	P2, P4, N1, N2, E1, E4

**Table 4 sensors-22-00067-t004:** Combinations of utterances that gave the maximum/minimum difference for each number of utterances (up to six utterances).

Number of Utterances	Combination of Utterances That Gives Maximum Difference	Combination of Utterances That Gives Minimum Difference
1	E3	E2
2	N1, E3	P4, E2
3	N1, E3, E4	N2, E2, E4
4	P3, N1, E3, E4	P4, N2, E2, E4
5	P3, P4, N1, E3, E4	P4, N1, N2, E1, E2
6	P3, P4, N1, E1, E3, E4	P3, P4, N1, N2, E1, E2

## Data Availability

The data presented in this study are available on request from the corresponding author. The data are not publicly available due to privacy and ethical restrictions.

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
