# Peer review of "Performance Evaluation of a Voice-Based Depression Assessment System Considering the Number and Type of Input Utterances"

_sensors, 2021, doi:10.3390/s22010067_

Round 1
Reviewer 1 Report
Mood changes affect facial expressions and speech sounds. In this paper, the authors attempt to assess people's depression level through speech attribute features, targeting brief speech input. The approach differs from the current approach using saliva, blood, etc., or using questionnaires, and has the advantage of being non-invasive and can be performed remotely.
The following are a few suggestions that, if further improved, could greatly improve the quality and persuasiveness of this paper.
In the methodology, it is suggested to further reveal the participants of the test, the recruitment conditions and the recruitment method; how to generate the reading of the 17 regular Japanese phrases; and what processing is done to ensure the reliability and validity of the test?
What is the theoretical basis or criterion for the correlation between the attribute features of speech sounds and mental states in Voice analysis and results? The significant differences between the experimental and control groups may have other potential extraneous variables that need to be further validated inferentially!
Author Response
Thank you very much for providing important comments. We are thankful for the time and energy you expended.
Point1: In the methodology, it is suggested to further reveal the participants of the test, the recruitment conditions and the recruitment method; how to generate the reading of the 17 regular Japanese phrases; and what processing is done to ensure the reliability and validity of the test?
Response 1: At H1, patients undergoing treatment for major depressive disorder were informed about our study at their first visit to the hospital. Then, 93 patients who gave informed consent were collected. In addition, 14 self-reported healthy volunteers (colleagues and university staffs) were recruited and their informed consent for our study was obtained. At H2, a total of 23 subjects (colleagues and connections) were recruited, all of them self-reported as healthy and their informed consent for our study was obtained. Then, we have replaced Table 1 in the previous manuscript with another table.
The 17 regular Japanese phrases were selected in consultation with a psychiatrist. In particular, the positive and negative phrases were chosen from the items that psychiatrists use in their daily interviews with patients.
Validation with other datasets is needed to ensure the reliability and validity of our results, which is a topic of future study.
Point 2: What is the theoretical basis or criterion for the correlation between the attribute features of speech sounds and mental states in Voice analysis and results? The significant differences between the experimental and control groups may have other potential extraneous variables that need to be further validated inferentially!
Response 2: We have added our previous studies to the reference. In addition, we have added the result of the correlation between HDRS score and vitality in depressed patients in order to show that vitality is a valid index to assess depression. Based on our previous studies and the correlation between HDRS score and standard vitality in depressed patients in this study, vitality was considered to be a valid index to assess depression. Since the number of severely depressed patients in this study was small, it was considered that the correlation between HDRS score and standard vitality in depressed patients was not sufficient.
The significant differences between the experimental and control groups may have potential factors other than the recording environment and age, however further analysis of the factors is not possible in this study and is a future study.
Reviewer 2 Report
Well-crafted contribution with high intrinsic merit. Technically sound. Th structure of the manuscript is well articulated. An in-depth discussion of the authors study is provided.
Author Response
Thank you very much for providing important comments. We are thankful for the time and energy you expended.
Reviewer 3 Report
The manuscript needs some improvement, as mentioned below.
Please, include the English version instead of romaji in table 2: I-ro-ha-ni-ho-he-to.
According to my opinion the sentence ”Today will be sunny” can be treated as the one with a positive impact. Therefore, I hardly agree it is an emotionless phrase. Please, comment on it. It might be the subject of the language layer.
The Authors should additionally make a comment on age of subjects, especially that standards deviation is of high values. It can influence the results. How? This is both a challenge of methodology - the papers is lack of any errors' computation. Therefore the methodology should be improved.
Author Response
Thank you very much for providing important comments. We are thankful for the time and energy you expended.
Point1: Please, include the English version instead of romaji in table 2: I-ro-ha-ni-ho-he-to.
Response 1: We have added "(Similar to “a-b-c”)."
Point 2: According to my opinion the sentence "Today will be sunny" can be treated as the one with a positive impact. Therefore, I hardly agree it is an emotionless phrase. Please, comment on it. It might be the subject of the language layer.
Response 2: "Today will be sunny" is a mistranslation; the correct translation is "It's fine today." So, we have modified that phrase. The meaning of E2 may have a positive impact, however since this phrase is used in Japan for microphone testing, we decided that E2 is not a positive word from a Japanese perspective. We have added a comment "Mic test phrase commonly used in Japan."
Point 3: The Authors should additionally make a comment on age of subjects, especially that standards deviation is of high values. It can influence the results. How? This is both a challenge of methodology - the papers is lack of any errors' computation. Therefore the methodology should be improved.
Response 3: We have performed an analysis of covariance with age as a covariate and added the results. For both the healthy and depressive groups, the standard vitality distributions were almost the same before and after adjustment for age. Therefore, age was not considered as a potential evaluation parameter for the experimental data. We recruited subjects from a wide range of age groups, including confirmation that there was no effect of age on the vitality. In fact, it was confirmed that the analysis result did not change with the standard vitality adjusted for age.